# Pathogenic Effect of *GDAP1* Gene Mutations in a Yeast Model

**DOI:** 10.3390/genes11030310

**Published:** 2020-03-14

**Authors:** Weronika Rzepnikowska, Joanna Kaminska, Dagmara Kabzińska, Andrzej Kochański

**Affiliations:** 1Neuromuscular Unit, Mossakowski Medical Research Centre Polish Academy of Sciences, 02-106 Warsaw, Poland; wrzepnikowska@imdik.pan.pl (W.R.); dagkab@imdik.pan.pl (D.K.); 2Institute of Biochemistry and Biophysics Polish Academy of Sciences, 02-106 Warsaw, Poland; kaminska@ibb.waw.pl

**Keywords:** yeast, mitochondria, Charcot-Marie-Tooth type 4 disease, *GDAP1* gene, GDAP1 gene mutations

## Abstract

The question of whether a newly identified sequence variant is truly a causative mutation is a central problem of modern clinical genetics. In the current era of massive sequencing, there is an urgent need to develop new tools for assessing the pathogenic effect of new sequence variants. In Charcot-Marie-Tooth disorders (CMT) with their extreme genetic heterogeneity and relatively homogenous clinical presentation, addressing the pathogenic effect of rare sequence variants within 80 CMT genes is extremely challenging. The presence of multiple rare sequence variants within a single CMT-affected patient makes selection for the strongest one, the truly causative mutation, a challenging issue. In the present study we propose a new yeast-based model to evaluate the pathogenic effect of rare sequence variants found within the one of the CMT-associated genes, *GDAP1*. In our approach, the wild-type and pathogenic variants of human *GDAP1* gene were expressed in yeast. Then, a growth rate and mitochondrial morphology and function of *GDAP1*-expressing strains were studied. Also, the mutant GDAP1 proteins localization and functionality were assessed in yeast. We have shown, that *GDAP1* was not only stably expressed but also functional in yeast cell, as it influenced morphology and function of mitochondria and altered the growth of a mutant yeast strain. What is more, the various *GDAP1* pathogenic sequence variants caused the specific for them effect in the tests we performed. Thus, the proposed model is suitable for validating the pathogenic effect of known *GDAP1* mutations and may be used for testing of unknown sequence variants found in CMT patients.

## 1. Introduction

In the past, molecular genetic testing was limited to single genes. Nowadays, whole exome sequencing (WES) is applied, the result of which is a series of DNA sequence variants even for the Mendelian diseases [1]. For example, in the hereditary peripheral neuropathies (Charcot-Marie-Tooth disorders; CMT) categorized as monogenic disorders (more than 80 genes involved), nearly 30% of the patients harbor two rare sequence and ten percent patients have three rare sequence variants in the CMT genes [2].

In Charcot-Marie-Tooth type 4A disease (CMT4A), causative mutations were identified in *GDAP1* (ganglioside induced differentiation associated protein 1) [3,4]. The role of the *GDAP1* gene in the physiology and pathology of the peripheral nerves is still not deciphered [5,6]. Recessive and dominant mutations within the *GDAP1* gene segregate with severe and moderate to mild clinical progressions of the disease, respectively [7,8].

It appears that the pathogenic effect of the *GDAP1* gene sequence variants reported until now varies substantially. However, different mutations were investigated in different manners. For some (causing Leu239Phe, Gly327Asp), the pathogenic effect has been documented by functional studies, whereas other characteristics are limited to clinical and electrophysiological observations. In addition, objective evaluation of the strength of any given mutation is hampered by the degree of mutation prevalence, which varies widely, i.e. some of the *GDAP1* gene mutations were reported in numerous CMT families (Glu163X, Ser194X), whereas others (His123Arg, Glu222Lys) were reported in small and even single pedigrees [6].

Initially, the *GDAP1* gene was identified as being induced upon enhanced ganglioside production [9]. Later it was shown that GDAP1 is a mitochondrial membrane protein mainly present in neuronal cells [10,11,12]. It is involved in mitochondria fission [13,14], calcium homeostasis [15,16], and maintenance of cellular redox potential [17,18], although the exact function of GDAP1 in these processes is unknown.

In yeast there is no simple homolog of human *GDAP1* gene, however it was shown that it complements some *FIS1* gene deletion phenotypes, coding for protein involved in mitochondrial fission process [19]. These results suggest that GDAP1 is functional in yeast cells and possesses molecular partners.

The aim of this study was to develop a reproducible yeast-based system for determination of the pathogenicity of *GDAP1* sequence variants. This system is based on the mitochondrial localization of GDAP1 protein and its mutations influence on mitochondria morphology and calcium signaling in mammalian cells. Here, we studied a localization of mutant GDAP1 proteins in yeast cells, its impact on mitochondrial network formation, the rate of mitochondrial DNA escape to the nucleus and the ability to grow on a non-fermentable carbon source. Additionally, we found that expression of wild-type *GDAP1* gene reduces the growth defects of the yeast *csg2*Δ mutant, with the deletion of a gene required for mannosylation of inositolphosphorylceramide, while mutated versions of *GDAP1* do not. In summary, our model may help to identify potentially pathogenic *GDAP1* alleles in CMT patients.

## 2. Materials and Methods

### 2.1. Strains, Media, and Growth Conditions

*Escherichia coli* strains DH5α and XL1-Blue were used for plasmid propagation. The yeast *Saccharomyces cerevisiae* strains used in this study were BY4741 MAT**a**
*his3*Δ*1 leu2*Δ*0 met15*Δ*0 ura3*Δ*0*, BY4741 *csg2*Δ and PTY44 MATα *ura3-52 lys2 leu2-3,112 trp1*Δ*1* [*rho+, TRP1*] [20].

Yeast were grown at 28 °C or 30 °C in YPD medium (1% yeast extract, 2% peptone, 2% glucose), YPG medium (1% yeast extract, 2% peptone, 3% glycerol) or YPG + 0.2% glucose, in minimal synthetic medium with glycerol (0.67% yeast nitrogen base with ammonium sulfate without amino acids, 3% glycerol with desired supplements (uracil, amino acids)) or in complete synthetic medium (SC) (0.67% yeast nitrogen base with ammonium sulfate without amino acids, 2% glucose or 3% glycerol with complete supplement mixture (CSM-ade-his-leu-trp-ura)) either solid or liquid. For growth tests, mitochondrial observations and mitochondrial DNA escape assays, a mixture of several yeast colonies, obtained after transformation, were grown overnight in liquid media. For growth tests the optical cell density (OD_600_) was determined and cultures were diluted with water to OD_600_ ∼1. Subsequently, aliquots of 10-fold serial dilutions of cells were spotted on solid media plates (as indicated) supplemented as indicated. Plates were incubated at 28 °C for the indicated number of days.

### 2.2. Plasmids

The plasmids used in this study are listed in Table 1.

The p425-P*_TDH3_*-*GDAP1* and p425-P*_TDH3_*-*GDAP1m1* plasmids were produced by amplification of full length or truncated *GDAP1* cDNA by PCR using a commercially available pCMV6-XL5 vector containing cDNA of *GDAP1* (NM_018972; OriGene) as a template and primers providing restriction sites (*Bam*HI at 5’-end and *Sal*I 3’-end) and additional codon stop for *GDAP1* m1 and ligation obtained alleles into p425-P*_TDH3_* vector. The m2 (c.980G>A; p.Gly327Asp), m3 (c.652C>G; p.Gln218Glu), m4 (c.664G>A; p.Glu222Lys), m5 (c.715C>T; p.Leu239Phe), m6 (c.368A>G; p.His123Arg), m7 (c.347T>C; p.Met116Thr), and m8 (c.467C>G; p.Ala156Gly) mutations were introduced by site directed mutagenesis into cDNA *GDAP1* gene.

### 2.3. Site Directed Mutagenesis

Site directed mutagenesis was performed on the OriGene plasmid: pCMV6-XL5 with *GDAP1* (NM_018972) Human Untagged Clone, using the Mut Express II Fast Mutagenesis Kit V2 (Vazyme) in accordance with the manufacturer’s instructions.

Primers for mutagenesis were designed using Quick Change Primer Design (Agilent) on-line software (the primer sequences were placed in Table A1).

*Escherichia coli* strain XL1-Blue was used for plasmid propagation. The presence of mutations within constructed plasmids was verified by the Sanger sequencing method.

### 2.4. Fluorescence Microscopy

Mitochondria morphology was observed in yeast cells grown overnight in SC-leu-his media at 30 °C and transferred into SC gly-leu-his for 4 h. The cells were viewed with an Eclipse E800 (Nikon, Tokyo, Japan) fluorescence microscope equipped with a DS-5Mc camera (Nikon). Images were collected using Lucia General 5.1 software (Laboratory Imaging Ltd., Prague, Czech Republic). The same fields were viewed by differential interference contrast (DIC) optics.

### 2.5. Confocal Microscopy

To observe the localization of GDAP1, yeast cells were grown overnight in SC-leu-his media at 28 °C and shifted into SC gly-leu-his for 4 h. Cells were fixed by incubation for 25 min in 3.7% formaldehyde. They were collected by centrifugation and washed four times in buffer B (40 mM potassium phosphate, pH 7.0; 0.5 mM MgCl2; 1.2 M sorbitol). Then they were resuspended in buffer B supplemented with 0.2 mg/mL Zymolyase 20T (Amsbio, Abingdon, UK) and 1% 2-mercaptoethanol and incubated for 1 h at 37 °C. The resulting spheroplasts were washed twice in buffer B and were spotted onto polylysine-coated slides. They were permeabilized by incubation in buffer F (20 mM potassium phosphate, pH 7.4; 150 mM NaCl; 0.1% BSA) supplemented with 0.1% Triton-X-100 for 15 min. The slides were rinse with buffer F and saturated in buffer F for 1 h. Then they were incubated with rabbit polyclonal anti-GDAP1 antibody (Abcam, Cambridge, MA, USA) diluted 1:200 in buffer F for 2 h. They were washed with buffer F and incubated with TRITC-conjugated porc anti-rabbit IgG antibody (DAKO, Agilent, Santa Clara, CA, USA) diluted 1:60 for 2 h. Then slides were rinsed with buffer F and nucleus was stained by DAPI (Thermo Fisher Scientific, Waltham, MA, USA) for 2 min. After being washed with water, slides were mounted in mounting medium (DAKO). Samples were viewed using LSM 780 Axio Observer Z.1 confocal microscope (Zeiss, Oberkochen, Germany). Images were collected using Zen 2012 black edition software (Zeiss). The confocal microscopy observations were performed in Laboratory of Advanced Microscopy Techniques, Mossakowski Medical Research Centre, PAS.

### 2.6. Western Blot Analysis

Yeast cells were grown overnight at 28 °C in SC -leu medium. Protein extracts were prepared after disrupting cells with acid-washed glass beads in 2 × urea electrophoresis sample buffer (50 mM Tris-HCl pH 6.8; 1.6% SDS; 7% glycerol; 0.016% bromophenol blue; 4% β-mercaptoethanol; 8M urea) supplemented with protease inhibitor cocktail (Sigma-Aldrich, Saint Louis, MI, USA). Samples were analyzed by standard Western blotting using rabbit polyclonal anti-GDAP1 (Abcam) or rabbit polyclonal anti-Histone H3 (Abcam) antibodies and secondary anti-rabbit IgG horseradish peroxidase (HRP)-conjugated antibodies (Sigma-Aldrich), followed by enhanced chemiluminescence (GE Healthcare, Boston, MA, USA).

### 2.7. DNA Escape Assay

DNA escape from mitochondria to the nucleus was assayed as in [20]. Briefly, the PTY44 strain was transformed with empty vector or one of the vectors bearing cDNA of *GDAP1* variants. The transformants were grown in minimal synthetic medium with glycerol-leu overnight and aliquots were spread on SC-trp or diluted and spread on YPD to count colony forming units. The percentage of cells able to grow on SC–trp was calculated for each type of transformants.

### 2.8. Estimation of Yeast Respiratory-Deficiency

Cells were grown overnight in YPG medium, diluted, and spread on a medium containing glycerol (non-fermentable) as the main source of carbon and with a limited amount of fermentable glucose (0.2%). We counted the fraction of small colonies (the respiratory deficient) whose growth was limited due to reduced levels of available fermentable glucose in relation to the total number of colonies.

## 3. Results

### 3.1. Selection of GDAP1 Mutations

From over 100 *GDAP1* gene sequence variants, we have selected the eight most representative for CMT4A disease in the Polish families diagnosed and observed for at least 5 years in the Warsaw Neuromuscular Unit (clinical electrophysiological and morphological studies have been previously reported). The main selection criterion was our detailed knowledge of the phenotypes associated with each mutation that have been extensively characterized in our laboratory [23,24,25,26]. We refer to our selected mutations as *GDAP1* m1 to m8 (Figure 1). Three mutations cause amino acid substitutions: Gly327Asp (m2), Leu239Phe (m5), and Met116Thr (m7), and belong to the class of mutations inherited as an autosomal recessive trait. The mutations Gly218Glu (m3), Ala156Gly (m8), and His123Arg (m6) represent pure dominant *GDAP1* gene mutations. It was shown that the *GDAP1* m4 mutation (Glu222Lys) is inherited in both dominant and recessive ways. The last mutation, that we called *GDAP1* m1, results in a C-terminal truncation of the GDAP1 protein (to 287 amino acid residues). This mutation is representative for the two recurrent nonsense *GDAP1* gene mutations i.e. Glu163X and Ser194X (Figure 1).

### 3.2. cDNA of the Human GDAP1 Gene Coding for Ganglioside Induced Differentiation Associated Protein 1 Is Stable When Heterologously Expressed in Yeast Cells

Due to the absence of homology between human *GDAP1* and any yeast gene we were unable to apply the strategy usually used in complementation tests. To address whether we can use yeast cells to assess the severity of the pathogenicity of *GDAP1* variants, we expressed the cDNA of the human wild-type *GDAP1* gene, and of the *GDAP1* variants, in yeast cells. First, we investigated the effect of *GDAP1* cDNA expression on growth of the wild-type yeast cells. Expression of neither the wild-type *GDAP1* nor any of the mutated variants (*GDAP1* m1–m8) affected the growth of yeast cultures on rich YPD (Figure 2A). To ensure that the *GDAP1* variants are properly expressed, the total cell extracts obtained from cells transformed with empty vector or bearing cDNA of *GDAP1* variants were prepared, separated by SDS-PAGE and immunoblotted with anti-GDAP1 antibodies. As a loading control, the levels of Histone 3 were monitored. Bands corresponding to GDAP1 and GDAP1 m1 proteins of appropriate molecular weights were observed (Figure 2B). The cDNAs of the missense *GDAP1* gene variants (*GDAP1* m1, m3, m4, m5, m6, m7, m8) were also stably expressed in wild-type yeast cells (Figure 2B). Thus, we were able to prove that in a heterologous system, human *GDAP1* variants are expressed and that their expression does not disturb the growth of wild-type yeast cell cultures. Due to the lack of any noticeable difference in growth between transformants expressing the cDNA of wild-type *GDAP1* or of its mutants, we had to search for conditions that allowed us to differentiate between cells producing wild-type GDAP1 and cells producing mutant GDAP1 proteins.

### 3.3. Specific GDAP1 Mutations Cause Mislocalization of the Resulting GDAP1 Protein

GDAP1 protein was been characterized as being localized to the mitochondrial membrane in neuronal cells [10,11,12]. Thus, in order to further characterize human GDAP1 functionality in the heterologous yeast-based system, we analyzed its cellular localization by confocal microscopy. The yeast cells were transformed with a plasmid encoding a mitochondrially targeted green fluorescence protein (mt-GFP) [22], empty vector or a series of *GDAP1* variants either with or without mutations. The wild-type GDAP1 protein was located in structures that matched the location of mt-GFP. Similarly, m3, m4, m5, m6 and m7 mutations did not result in changes in the subcellular localization of these GDAP1 protein variants. Two mutations, namely m1 and m2 (causing a C-terminal deletion and the Gly327Asp substitution, respectively), resulted in GDAP1 proteins that did not localize to mitochondria (Figure 2C and Figure A1). However, the level of fluorescence signal from the GDAP1 m2 protein was only slightly higher than the background signal observed for the empty vector control making it very difficult to observe the protein. The mislocalization of the GDAP1 m1 and m2 protein variants is most probably the result of an absence (in the case of GDAP1 m1) or functional impairment (for GDAP1 m2) of the transmembrane domain (TMD), which has previously been shown to be responsible for the mitochondrial membrane anchoring of GDAP1. Similarly to previous observations made in COS7 cells, mutations that resulted in changes within the TMD (e.g. GDAP1 m2) of the GDAP1 protein impaired the targeting of GDAP1 to the mitochondrial membrane [25].

### 3.4. Expression of Human GDAP1 cDNA Affects Mitochondrial Network Formation in Yeast

In previous studies, GDAP1 has been shown to alter the formation of mitochondrial networks when expressed in HeLa cells [23]. In our study we investigated whether expression of *GDAP1* in the heterologous system also changes the morphology of mitochondrial networks, especially given that the human protein is also localized to the mitochondria. The mitochondria were visualized by expression of mt-GFP. The network of mitochondria of cells transformed with empty vector were used as a reference. As expected, mitochondria formed tubular network structures below the cell cortex. After the introduction of cDNA encoding human GDAP1 protein, we observed an alteration of the mitochondrial network. We named this a supernetwork due to the more extensive and branched nature of the network (Figure 3A). Compared to the wild–type yeast, about 50% of the counted *GDAP1* transformed cells displayed this supernetwork phenotype. We also observed spherical and abnormal phenotypes, but there were no marked differences in the number of cells with such mitochondria between the cells expressing the cDNA of wild-type *GDAP1* and those that were not (Figure 3A).

We have confirmed the role of human GDAP1 protein in the formation of the mitochondrial network in yeast cells. The phenotypes we observed, however, were not suitable to distinguish between GDAP1 mutants or to use as a prognostic marker to assess pathogenicity of *GDAP1* mutations.

### 3.5. Some GDAP1 Protein Mutants When Produced in Yeast Increase the Rate of Mitochondrial DNA Escape to the Nucleus

In the previous experiment we have shown that expression of the cDNA from the human *GDAP1* gene influences the formation of the mitochondrial network in yeast. Moreover, a relationship between mitochondrial network disturbances and mutations in the *GDAP1* gene was reported [12,14]. Taking into account that the mitochondrial network formation is directly associated with mitochondrial biogenesis and that this process is accompanied by an exchange of mitochondrial DNA [27,28], we decided to check the functionality of mitochondria by assessment of mtDNA escape to the nucleus [29]. Thus, we hypothesized that at least some *GDAP1* gene mutations may affect mitochondrial DNA escape; from the mitochondria to the nucleus. To measure the rate of mtDNA escape to the nucleus, the *TRP1* gene, encoding an enzyme from the tryptophan biosynthesis pathway, was integrated into the mitochondrial genome [20,30] in a strain carrying a defective *trp1* allele in the (nuclear) genome. Such yeast is unable to grow without tryptophan supplementation. During the propagation of such yeast, mtDNA escapes from the mitochondria to the nucleus where it integrates and complements the defective nuclear *TRP1* allele allowing cells to grow on the media without tryptophan [20,31].

The idea of this experiment may be summarized by the following simple association: more extensive abnormal mitochondria leads to a greater rate of DNA escape and reflects the severity of the *GDAP1* gene mutation. The greatest mitochondrial DNA escape rate was observed when the cDNA of *GDAP1* m5 (resulting in the Leu239Phe substitution) was expressed (Figure 3B). A moderate effect was observed for *GDAP1* m2 (Gly327Asp substitution within the TMD). The effect of expressing other *GDAP1* variants was not strong. There were no statistically relevant differences in mtDNA escape rates between cells bearing empty plasmid or cDNA of the *GDAP1* m1, m3, m4, m6, m7, and m8 variants. Altogether, these results show that although the expression of *GDAP1* does not affect the growth of yeast cells, it does affect their physiology, in this instance the mitochondria. The changes are so extensive that they can be observed. This supports the idea of using a yeast model to study pathologies caused by mutations in the *GDAP1* gene.

An altered morphology of mitochondria is often associated with a defective respiratory function [32]. We asked whether the observed changes in the morphology of mitochondria translate into the ability to grow on a non-fermentable carbon source, the conditions requiring functional mitochondrial respiration. The expression of wild type *GDAP1* caused a significant increase in the number of cells which are unable to grow on non-fermentable carbon source containing media (*p* < 0.05) and thus are respiratory deficient (Figure 3C). This effect was significantly reduced when *GDAP1* m1, m2, m6, and m7 variants were expressed (*p* < 0,05) and was not altered by *GDAP1* m3, m4, and m5 variants (Figure 3C). The decrease in the number of respiratory deficient cells between yeast transformants expressing different pathogenic *GDAP1* variants and the wild type *GDAP1* reflects the influence of mutations on GDAP1 protein function connected with the observed phenotype.

### 3.6. The Expression of Human Wild-Type GDAP1 Gene Reduces the Growth Defect of the csg2 Mutant Yeast Strain

The yeast-based model of the CMT4A disease that is obtained through the heterologous expression of the cDNA of *GDAP1* and its eight mutants represents a stable and functional system which has several limitations. The main limitation lies in the mutation-specific phenotype. In essence, only some *GDAP1* mutations may be analyzed using the mtDNA escape assay. Similarly, the phenotypes based upon colocalization of the GDAP1 with mitochondria are only suitable to distinguish whether mutations affect the transmembrane domain of GDAP1. To obtain a simple model to use as a potential platform for further drug screening, we looked for a repeatable growth phenotype.

Based on GDAP1 localization data obtained from the literature, on the abnormalities described as being associated with *GDAP1* mutations and on data from studies of other genes in which mutations also cause CMT disease, we expressed the cDNA of *GDAP1* in several specifically selected yeast deletion mutants. We assumed that we should be able to find a mutant and/or conditions which allow us to monitor *GDAP1* expression in simple growth test. One of our candidates was *csg2*Δ mutant, with the deletion of a gene required for mannosylation of inositolphosphorylceramide. This mutant has higher levels of reactive oxygen species (ROS) and loss of mitochondrial DNA suppress this accumulation [33]. Thus, we suspected that expression of *GDAP1* or its variants which results accumulation of Rho^-^ cells could be helpful for growth of *csg2*Δ at stress conditions. We found that the tunicamycin and calcium ion (Ca^2+^) sensitivity of the *csg2*Δ strain is suppressed by expression of wild-type the *GDAP1* allele (Figure 4A). The truncated version of GDAP1 protein (GDAP1 m1) and the mutation in which localization to the mitochondria is perturbed (GDAP1 m2) were not able to restore the normal growth of mutant cells on either plates containing tunicamycin or Ca^2+^ ions. Similar to *GDAP1* m1, the expression of *GDAP1* m3 was unable to restore growth of the *csg2*Δ cells in either test condition. The expression of *GDAP1* m5, m6, and m7 restored the growth of the *csg2*Δ mutant to a lesser degree; weaker when compared to the wild-type *GDAP1* but stronger than *GDAP1* m1. In these terms the mutations *GDAP1* m1, m2, m3, m5, m6, m7, and m8 could be classified as loss-of-function or partial loss-of-function mutations. Interestingly in this assay the effect of the Glu222Lys substitution caused by the GDAP1 m4 mutation was negligible and the growth of the *csg2*Δ mutant expressing *GDAP1* m4 was fully restored on tunicamycin and restored to a slightly lesser extent on Ca^2+^ ion containing medium, in comparison to wild-type *GDAP1* expression (Figure 4A). The lack of complementation by some *GDAP1* variants is not due to lack of protein expression as shown by Western blot analysis of GDAP1 proteins (Figure 4B). In summary, our study has identified a new phenotype connected with expression of the *GDAP1* cDNA in yeast cells. This supported our hypothesis that we should be able to identify conditions under which the presence of GDAP1 protein can be studied in a simple growth assay. This test may help to identify at least some of the loss-of-function alleles found in patients.

## 4. Discussion

One of the main problems faced by modern medical genetics is the proper identification of a causative mutation followed by determination of the degree of pathogenicity in order to predict the progress of the disease. Furthermore, most genetic disorders are rare, making them very difficult to understand and ultimately to find treatment. It is also extremely expensive to develop efficient treatments for such diseases. These problems also apply to the vast majority of the neurodegenerative heritable disorders. Studies of therapies for neurodegenerative disorders are underdeveloped, even on the preclinical level. In the present study we used, for the first time, a yeast-based model to establish a system in which to assess the pathogenic effect of *GDAP1* gene mutations.

More than 15 years of studies have been devoted to the analysis of *GDAP1* mutations. There are some well documented variants such as the *GDAP1* m5 (Leu239Phe) mutation reported in numerous patients. However, several mutations are poorly documented. For example, the *GDAP1* m6 (His123Arg) mutation was found only in two families. Thus, testing of pathogenicity is even more necessary in instances such as the *GDAP1* m6 variant [23,24]. Even more interesting, and underexplored is the pathogenic status of the *GDAP1* m4 (Glu222Lys) mutation. The *GDAP1* m4 mutation has been shown by us to segregate with an extremely mild phenotype in the case of autosomal dominant CMT but in combination with the *GDAP1* m5 (Leu239Phe) mutation it causes severe CMT with autosomal recessive traits [26]. Interestingly, out of the mutations with no clear phenotype in the yeast growth reduction assay, it was the recurrent and pathogenic *GDAP1* m5 mutation that resulted in the highest observed level of mtDNA escape. This may indicate a separate and specific mechanism of action for the *GDAP1* m5 mutation. The *GDAP1* m8 (Ala156Gly) mutation, which segregates with a CMT with dominant inheritance traits and a moderate clinical pathology, also results in a moderate pathogenic effect in our yeast growth assay. In our study, the *GDAP1* m4 (Glu222Lys) mutation has the lowest pathogenic effect (Table 2). Although our previous results showed a clear segregation of the *GDAP1* m4 variant with two CMT phenotypes, its presence has only been documented in two families. There is still a possibility that considering the small group of samples, the association of the *GDAP1* m4 variant with the CMT4A phenotype may be random. Thus, we cannot exclude the possibility that the *GDAP1* m4 variant may be categorized as a harmless polymorphism.

Finally, our study confirms hypothesis that the *GDAP1* mutations located within the transmembrane domain result the severest phenotypes. These mutations lead to the mislocalization of the GDAP1 protein to the cytoplasm instead of to the mitochondrial membrane [25]. In the present study we have shown these types of *GDAP1* gene mutations are also associated with the strongest pathogenic effect. This observation is in accordance with previous clinical studies reporting extremely severe phenotypes associated with Glu163X, Ser194X, Thr288fsx290, and Gly327Asp *GDAP1* gene mutations. The patients harboring these mutations become wheelchair-dependent during infancy or early adulthood [8,25]. We propose that because the localization of the GDAP1 protein to the mitochondrial membrane is critical for its function, the testing of GDAP1 location should be the first step in diagnosing the severity of a mutation, especially for mutations affecting the amino acid sequence of the TMD. Furthermore, due to the observed heterogeneity in the mechanisms by which *GDAP1* mutations act, the tests used to assess their pathogenicity should be based upon the biological properties/parameters summarized in Table 2.

The model established here is shown to be suitable for validating the pathogenic effect of known *GDAP1* mutations. Our system seems to be particularly suited for the validation of new sequence *GDAP1* variants identified in single patients manifesting with CMT. The ability to test in a simple yeast-based model is attractive due to its ease of use and low cost. Even when the CMT phenotype has been narrowed by careful clinical and electrophysiological assessment, the whole exome sequencing approach (WES) may generate an average of 2 or 3 variants within CMT genes with ambiguous pathogenic effect [2]. In this instance, access to a reproducible platform for functional studies may be a necessary and useful tool for assessing the causative role of a certain CMT sequence variant.

## Figures and Tables

**Figure 1 genes-11-00310-f001:**
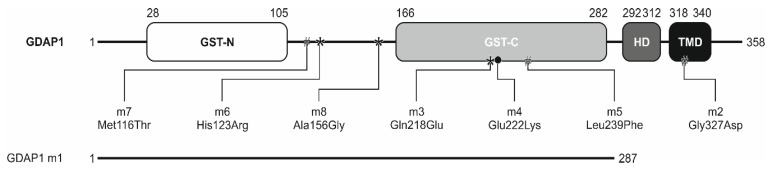
Schematic representation of the GDAP1 domain structure. Studied protein changes are indicated. GDAP1 m1 is a truncated version of GDAP1, lacking the domains responsible for correct localization of the protein (HD and TMD). Dominant mutations leading to the given substitution are flagged by stars (*) and recessive instances by a hashtag (#). *GDAP1* m4, which may be inherited in both dominant and recessive ways, is indicated by dot (•). Domains (based on Pfam database): (GST) glutathione S-transferase domain; (HD) hydrophobic domain; (TMD) transmembrane domain.

**Figure 2 genes-11-00310-f002:**
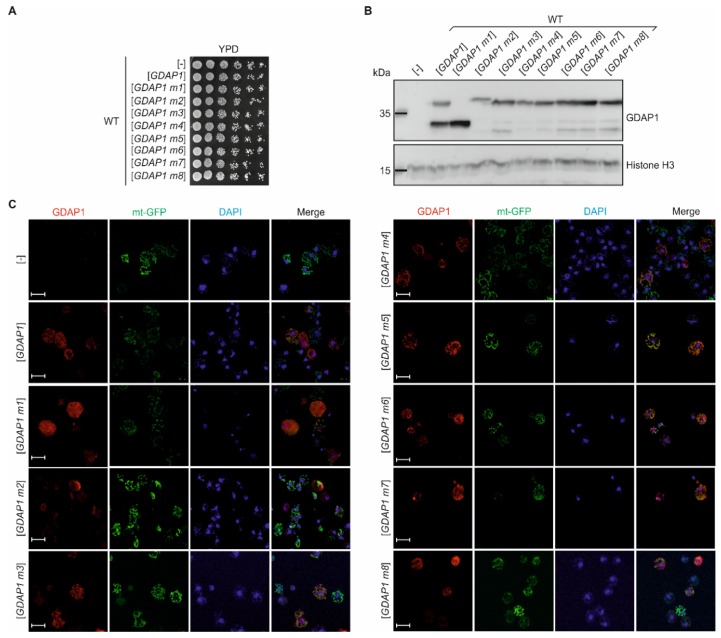
Yeast-based system to analyze GDAP1 protein function. (**A**) Overnight cultures of wild-type yeast cells harboring empty vector ([-]) or plasmids with *GDAP1* gene variants were diluted to OD_600_ ≈ 1 and then ten-fold serial dilutions were prepared and spotted on to YPD medium. Plates were incubated for 2 d at 28 °C. (**B**) Yeast wild-type cells harboring empty vector ([-]) or plasmids with *GDAP1* gene variants (as indicated) were cultured overnight in SC-leu at 28 °C, then were collected and disrupted. The total cell extracts obtained were analyzed by SDS-PAGE, followed by immunoblotting with anti-GDAP1 or anti-Histone H3 antibodies, as indicated. (**C**) Yeast wild-type cells transformed with plasmid encoding mitochondrially targeted green fluorescent protein (mt-GFP) and with empty vector ([-]) or plasmids encoding *GDAP1* alleles (as indicated) were grown in SC–leu-his medium overnight and then were shifted to glycerol-containing medium for 4 h to induce expansion of mitochondria. Cells were fixed, stained and observed using confocal microscopy. GDAP1 was visualized by immunostaining (red), mitochondria using the mt-GFP protein marker (green) and the nucleus was stained with DAPI (blue). Scale bar 5 µm.

**Figure 3 genes-11-00310-f003:**
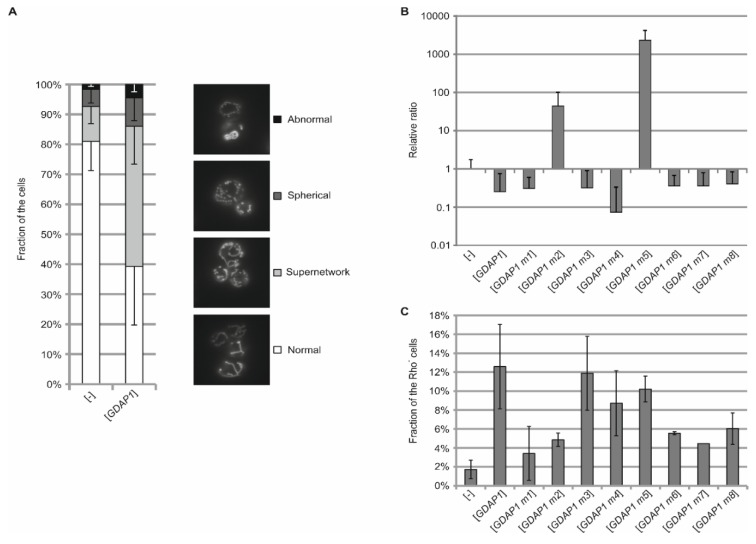
Mitochondria functioning is changed upon *GDAP1* expression in yeast. (**A**) Overnight cultures of wild-type yeast cells harboring plasmid bearing mitochondrially targeted green fluorescent protein (mt-GFP) and empty vector ([-]) or plasmid with full-length *GDAP1* were shifted to glycerol-containing medium for 4 h to induce expansion of mitochondria. Mitochondrial morphology was observed using fluorescent microscopy. The described phenotypes were counted. Error bars represent standard deviation for three independent biological experiments. (**B**) PTY44 strain transformed with empty vector ([-]) or *GDAP1* gene variants (as indicated) was incubated for 2 days in glycerol-containing medium and then the cells were counted, and dilutions of every culture were plated on YPD or SC-trp medium. After 3 d of incubation at 28 °C the number of colony-forming units (CFUs) was calculated. The fraction of SC-trp-growing cells in every culture were normalized to PTY44 with empty vector ([-]). Error bars represent standard deviation for four experiments. (**C**) Wild-type yeast strain harboring empty vectors ([-]) or plasmids with *GDAP1* alleles were grown overnight in SC gly-leu medium. The cultures were diluted, and cells were plated on YPG + 0.2% glucose and incubated for 3 d at 28 °C. CFUs were counted and the percentage of small colonies was calculated. Error bars represent standard deviation for four experiments.

**Figure 4 genes-11-00310-f004:**
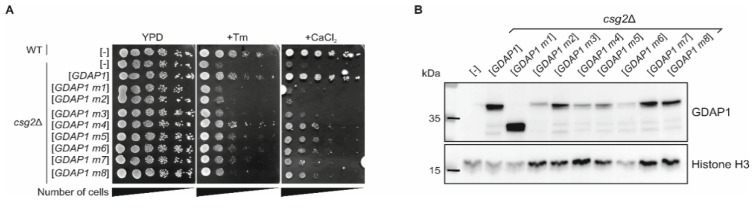
Possible yeast-based model of CMT4A disease for drug screen. (**A**) Overnight cultures of wild-type or *csg2*Δ yeast cells harboring empty vector ([-]) or plasmids with *GDAP1* alleles were diluted to OD600 ≈ 1. Ten-fold serial dilutions were prepared and spotted on YPD media containing 0.5 µg/mL tunicamycin (Tm) or 0.5 M calcium chloride (CaCl_2_). Plates were incubated for 2 d (YPD and + Tm) or 6 d (+ CaCl_2_) at 28 °C. (**B**) The *csg2*Δ yeast strain harboring empty vector ([-]) or plasmids with *GDAP1* alleles (as indicated) were cultured overnight in SC-leu medium at 28 °C. Cells were then harvested and disrupted. Total cell extracts were analyzed by SDS-PAGE, followed by immunoblotting with anti-GDAP1 or anti-Histone H3 antibodies. Histone H3 was used as a loading control.

**Table 1 genes-11-00310-t001:** Plasmid used in this study.

Plasmid	Source or Reference
pCMV6-XL5-*GDAP1*	OriGene
p425-P*_TDH3_* [2µ; *LEU2*]	[21]
p425-P*_TDH3_*-*GDAP1*	This study
p425-P*_TDH3_*-*GDAP1m1*	This study
p425-P*_TDH3_*-*GDAP1m2*	This study
p425-P*_TDH3_*-*GDAP1m3*	This study
p425-P*_TDH3_*-*GDAP1m4*	This study
p425-P*_TDH3_*-*GDAP1m5*	This study
p425-P*_TDH3_*-*GDAP1m6*	This study
p425-P*_TDH3_*-*GDAP1m7*	This study
p425-P*_TDH3_*-*GDAP1m8*	This study
PXY122 (mt-GFP) [2µ, *HIS3*]	[22]

**Table 2 genes-11-00310-t002:** Pathogenic effect of *GDAP1* gene mutations.

*GDAP1* Variant (aa Substitution)	Localization	Inheritance of *GDAP1* Mutation	Petite	mtDNA Escape	Effect on *csg2*Δ		Summarized Effect in Yeast	Clinical Phenotype
					Tm	CaCl_2_		
*GDAP1*	M	_	12	-	+	+	_	_
*GDAP1 m1* (ΔC-terminus)	C	AR	4	-	-	-	strong	severe
*GDAP1 m*2 (Gly327Asp)	C	AR	5	moderate	-	-	strong	severe
*GDAP1 m*3 (Glu218Glu)	M	AR	11	-	-	-	strong/ moderate	moderate to severe
*GDAP1 m*4 (Glu222Lys)	M	AR/AD	9	minimal	+	+	weak	weak to moderate
*GDAP1 m*5 (Leu239Phe)	M	AR	10	maximal	-/+	-/+	moderate	moderate in homozygote, lack in heterozygote
*GDAP1 m*6 (His123Arg)	M	AD	5	-	-/+	-/+	weak	weak
*GDAP1 m*7 (Met116Thr)	M	AR	5	-	-/+	-/+	weak	moderate in homozygote
*GDAP1 m*8 (Ala156Gly)	M	AD	6	-	-	-/+	moderate	moderate in heterozygote

The summarized results illustrating the pathogenic effect of *GDAP1* versions analyzed on selected phenotypes in yeast in comparison to the clinical effect. Localization: C-cytoplasmic, M-mitochondrial; Inheritance AR—autosomal recessive, AD—autosomal dominant; Petite—the results show the percentage of small colonies in the cells expressing *GDAP1* variants; mt DNA escape —the effect of the *GDAP1* variants expression on a rate of mitochondrial DNA escape reported as maximal, moderate or minimal; effect of *GDAP1* variants expression on the *csg2*Δ mutant ability to grow on media containing tunicamycin (Tm) or calcium chloride (CaCl_2_) (+) strong, (+/-) moderate, or (-) no effect.

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
