# Peer review of "Pathogenic Effect of GDAP1 Gene Mutations in a Yeast Model"

_genes, 2020, doi:10.3390/genes11030310_

Round 1
Reviewer 1 Report
This study successfully used a yeast model to investigate the function of gene variants of the human ganglioside induced differentiation associated protein 1 (GDAP1). Site directed mutagenesis was used to create eight mutants in this gene which were previously identified and representative of those with CMT4A disease in Warsaw, Poland. Although yeast does not have a GDAP1 ortholog to complement directly, previous work has shown that expression of GDAP1 in yeast can complement phenotypes of yeast strains defective for genes involved in mitochondrial fission or fusion.
The authors find that the cDNA for the human GDAP1 gene is stably produced in yeast and properly sized protein is produced as evidenced by Western analysis. Microscopic analysis of GDAP1 mutants expressed in yeast are shown in Fig. 2 C. It is difficult to see these images and this figure would be enhanced by enlarging the panels to allow the readers to see the mitochondrial phenotypes. Perhaps add larger panels in the supplemental materials for the reader to better see the phenotypes. Also, growing in glycerol for 4 hours to induce the expansion of the mitochondrial phenotype is rather short. It would be worth looking at some of these phenotypes after a longer growth period in glycerol to enhance the phenotype given the efforts the authors have performed to use immunostaining to detect the protein in fixed preparations (perhaps not for this study but for future work).
The microscopic work presented in Fig. 2C allowed the authors to identified the mislocalization of GDAP1 in yeast related to the specific mutants generated in the gene compared to the wildtype GDAP1. However, the phenotypes presented did allow the authors to distinguish between specific mutants. This reviewer wonders if the yeast cells were grown longer in glycerol to further expand the mitochondrial network, it might have been possible to resolve differences in the phenotype of the specific mutants (if present).
The authors should also better introduce the mitochondrial DNA escape assay earlier in the paper to help the reader understand why it is being used in this work. Why are these specific GDAP1 mutants causing the generation of mitochondrial damage which leads to mtDNA escape? Why, for example does the GDAP1 m5 variant lead to such a large amount of mtDNA escape (mitochondrial damage) relative to the empty vector but this is not evident from the microscopic evidence? Does this have something to do with the extended culture in glycerol media? Why are the error bars for m2 and m5 so small compared to the other variants in Fig. 3B. Is there something about these mutants which stabilizes the phenotype to such an extent that the variation is limited?
Why was the csg2 knockout initially selected to be included in this work? Reference to prior work associated with this gene and mitochondrial function/interaction with GDPA1 or was this identified from a previous screen?
There appears to be a bubble or some factor that interferes with the detection of GDAP1 m6 and m7 proteins in Fig. 4B.
The authors did a nice job of showing that mutations in the transmembrane domain result the most extreme phenotypes.
Also, suggestion. It would be worth performing in future work a synthetic genetic array (SGA) experiment to investigate the interaction of GDAP1 expression in yeast against the yeast knockout library, the essential library, and the overexpression library to identify other yeast genes that could modify the respiration/mitochondrial function of yeast cells. That approach may allow the authors to discover additional yeast mutants for testing GDAP1 in this model organism.
Author Response
We highly appreciate all efforts of the reviewer to improve our manuscript, both in general and detailed points. We would like to thank for all advices and suggestions. We have addressed all the points indicated by the reviewer as follows.
- The introduction must be improved
We have improved the introduction paragraph including the function of GDAP1 protein and describing in more details our yeast-based system [lines 50-68: Initially, the GDAP1 gene was identified as being induced upon enhanced ganglioside production [9]. Later it was shown that GDAP1 is a mitochondrial membrane protein mainly present in neuronal cells [10–12]. It is involved in mitochondria fission [13,14], calcium homeostasis [15,16] and maintenance of cellular redox potential [17,18], although the exact function of GDAP1 in these processes is unknown.
In yeast there is no simple homolog of human GDAP1 gene, however it was shown that it complements some FIS1 gene deletion phenotypes, coding for protein involved in mitochondrial fission process [19]. These results suggest that GDAP1 is functional in yeast cells and possesses molecular partners.
The aim of this study was to develop a reproducible yeast-based system for determination of the pathogenicity of GDAP1 sequence variants. This system is based on the mitochondrial localization of GDAP1 protein and its mutations influence on mitochondria morphology and calcium signaling in HeLa cells. Here we studied a localization of mutant GDAP1 proteins in yeast cells, its impact on mitochondrial network formation, the rate of mitochondrial DNA escape to the nucleus and the ability to grow on a non-fermentable carbon source. Additionally we found that expression of wild-type GDAP1 gene reduces the growth defects of the yeast csg2Δ mutant, with the deletion of a gene required for mannosylation of inositolphosphorylceramide, while mutated versions of GDAP1 do not. In summary our model may help to identify potentially pathogenic GDAP1 alleles in CMT patients.]
-
Microscopic analysis of GDAP1 mutants expressed in yeast are shown in Fig. 2 C. It is difficult to see these images and this figure would be enhanced by enlarging the panels to allow the readers to see the mitochondrial phenotypes. Perhaps add larger panels in the supplemental materials for the reader to better see the phenotypes. Also, growing in glycerol for 4 hours to induce the expansion of the mitochondrial phenotype is rather short. It would be worth looking at some of these phenotypes after a longer growth period in glycerol to enhance the phenotype given the efforts the authors have performed to use immunostaining to detect the protein in fixed preparations (perhaps not for this study but for future work).
The microscopic work presented in Fig. 2C allowed the authors to identified the mislocalization of GDAP1 in yeast related to the specific mutants generated in the gene compared to the wildtype GDAP1. However, the phenotypes presented did allow the authors to distinguish between specific mutants. This reviewer wonders if the yeast cells were grown longer in glycerol to further expand the mitochondrial network, it might have been possible to resolve differences in the phenotype of the specific mutants (if present).
The enlarged panel for Figure 2C was added to the supplementary materials (Figure A1). We decided to 4 hours shift to glycerol to allow cells to one division. We expected that the differences between particular mutants would be visible just at the beginning, especially that we observed changes in mitochondrial morphology in GDAP1-expressing cells after this time (Figure 3A). We were afraid that longer growth in glycerol-containing medium allows cells to compensate some negative effects and eliminate the phenotypes resulted from expression of mutated alleles, as we observed for GDAP1-expressing cells after overnight culture in glycerol-containing medium. However in the future work we will take into account the advice of the reviewer and test phenotypes after longer growth period in the glycerol.
-
The authors should also better introduce the mitochondrial DNA escape assay earlier in the paper to help the reader understand why it is being used in this work. Why are these specific GDAP1 mutants causing the generation of mitochondrial damage which leads to mtDNA escape? Why, for example does the GDAP1 m5 variant lead to such a large amount of mtDNA escape (mitochondrial damage) relative to the empty vector but this is not evident from the microscopic evidence? Does this have something to do with the extended culture in glycerol media? Why are the error bars for m2 and m5 so small compared to the other variants in Fig. 3B. Is there something about these mutants which stabilizes the phenotype to such an extent that the variation is limited?
In the paragraph 3.5. we further justified the use of mtDNA escape method. [lines 259-267: In the previous experiment we have shown that expression of the cDNA from the human GDAP1 gene influences the formation of the mitochondrial network in yeast. Moreover, relationship between mitochondrial network disturbances and mutations in the GDAP1 gene was reported [14,27]. Taking into account that the mitochondrial network formations is directly associated with mitochondrial biogenesis and that this process is accompanied by an exchange of mitochondrial DNA [29,30], we decided to check the functionality of mitochondria by assessment of mtDNA escape to the nucleus]. Increased rate of the mtDNA escape cannot be simply extrapolated to mitochondrial morphology. For example GDAP1 clearly changes the mitochondrial network presentation but not alter or even decrease the rate of mtDNA escape to the nucleus. Data presented so far suggest that multiple pathways exist for the transfer of mtDNA to the nucleus. Thus it is possible that different mutations influence different cellular pathways to induce or not this phenomenon. Standard deviations for m2 and m5 are similar or even higher compare to other mutations ([-] 1,47911E-06; [GDAP1] 9,84525E-07; [m1] 5,77049E-07; [m2] 0,000113928; [m3] 1,1542E-06; [m4] 5,13345E-07; [m5] 0,00370879; [m6] 6,23076E-07; [m7] 8,5908E-07; [m8] 8,48776E-07). The error bars for these mutants were corrected.
-
Why was the csg2 knockout initially selected to be included in this work? Reference to prior work associated with this gene and mitochondrial function/interaction with GDPA1 or was this identified from a previous screen?
The csg2Δ mutant was included to the study based on the facts that: 1. the GDAP1 gene was found as being induced upon enhanced ganglioside (derivate of glycosphingolipids) production 2. this mutant has higher level of Reactive oxygen species (ROS) (similarly to GDAP1 knock-out cells) and 3. loss of mitochondrial DNA suppress this accumulation. For clarity we included explanation to the text [line 309-314].
-
There appears to be a bubble or some factor that interferes with the detection of GDAP1 m6 and m7 proteins in Fig. 4B.
As it was suggested we have replaced the panel 4b for a new one.
-
Also, suggestion. It would be worth performing in future work a synthetic genetic array (SGA) experiment to investigate the interaction of GDAP1 expression in yeast against the yeast knockout library, the essential library, and the overexpression library to identify other yeast genes that could modify the respiration/mitochondrial function of yeast cells. That approach may allow the authors to discover additional yeast mutants for testing GDAP1 in this model organism.
Suggested experiments are very interesting and valuable. No doubt, we included them for our future work.
Reviewer 2 Report
In this article titled “Pathogenic effect of GDAP1 gene mutations in a yeast model”, the authors have characterized phenotype of yeast strains expressing wild type and mutant versions of the heterologous gene GDAP1. The article generally reads well, provides some interesting insights into the effect of mutant GDAP1 expression in yeast cells and would be of interest to researchers in the field. However, there are few deficiencies which needs attention:
- In materials and methods section, the authors mention (lines 59-60) “minimal SM or complete SC (0.67% yeast nitrogen base…..”. I am not sure what the authors mean by “SM” medium? Is it a typo for CM (complete synthetic medium) or SD (standard minimal medium)? Also, it is not mentioned whether their media contain ammonium sulfate (a common ingredient of minimal media).
- Line 64: “cultures were diluted with water to OD600 ∼ 1”. It is well known that sudden change of media with water can result in osmotic/nutrient shock in yeast cells with many consequences. Since water was used to dilute the cultures, could some of the observations be related to this kind of stress? Ideally, cells should be diluted in appropriate media to minimize such shock.
- Line 107: “rinses” should be “rinsed”
- Lines 310-311: “the pathogenicity of variants found in patients can be tested in a simple growth assay”. Phenotypic observations made in yeast resulting from heterologous expression of a mutant human gene does not necessarily reflect pathogenicity observed in human cells (due to the large differences in functioning/cellular pathways between yeast and human). As such, this sentence appears to be an overstatement. This needs to be toned down a bit and needs to be discussed/clarified better.
Author Response
We highly appreciate all efforts of the reviewer to improve our manuscript, both in general and detailed points. We would like to thank for all advices and suggestions. We have addressed all the points indicated by the reviewer as follow.
1. In materials and methods section, the authors mention (lines 59-60) “minimal SM or complete SC (0.67% yeast nitrogen base…..”. I am not sure what the authors mean by “SM” medium? Is it a typo for CM (complete synthetic medium) or SD (standard minimal medium)? Also, it is not mentioned whether their media contain ammonium sulfate (a common ingredient of minimal media).
The description of the media was improved. We also added the information about ammonium sulfate contain [lines 74-81: Yeast were grown at 28°C or 30°C in YPD medium (1% yeast extract, 2% peptone, 2% glucose), YPG medium (1% yeast extract, 2% peptone, 3% glycerol) or YPG + 0.2% glucose, in minimal synthetic medium with glycerol (0.67% yeast nitrogen base with ammonium sulfate without amino acids, 3% glycerol with desired supplements (uracil, amino acids)) or in complete synthetic medium (SC) (0.67% yeast nitrogen base with ammonium sulfate without amino acids, 2% glucose or 3% glycerol) with complete supplement mixture (CSM –ade –his –leu –trp -ura)) either solid or liquid.]
2. Line 64: “cultures were diluted with water to OD600 ∼ 1”. It is well known that sudden change of media with water can result in osmotic/nutrient shock in yeast cells with many consequences. Since water was used to dilute the cultures, could some of the observations be related to this kind of stress? Ideally, cells should be diluted in appropriate media to minimize such shock.
It is good point that change of media with water can result in osmotic/nutrient shock in yeast cells. However cells stayed in the water only for a short period of time, needed for dropping them on the plates with appropriate media. Moreover cells were growth on media for several days, thus the effect of osmotic shock is rather minimal if at all. Additionally all controls were treated in the same way and phenotypes observed for csg2Δ strain are consistent with those found in literature.
3. Line 107: “rinses” should be “rinsed”
The word has been changed as indicated.
4.Lines 310-311: “the pathogenicity of variants found in patients can be tested in a simple growth assay”. Phenotypic observations made in yeast resulting from heterologous expression of a mutant human gene does not necessarily reflect pathogenicity observed in human cells (due to the large differences in functioning/cellular pathways between yeast and human). As such, this sentence appears to be an overstatement. This needs to be toned down a bit and needs to be discussed/clarified better.
We have changed indicated sentence as follow: [lines 330-333] This supported our hypothesis that we should be able to identify conditions under which the presence of GDAP1 protein can be studied in a simple growth assay. This test may help to identify, at least some of the loss-of-function alleles found in patients.
Round 2
Reviewer 2 Report
Thank you for addressing the points raised by me for the previous version of the manuscript. Wish you all good luck for the future!